# Role for gene conversion in the evolution of cell-surface antigens of the malaria parasite *Plasmodium falciparum*

**Brice Letcher** [1,2] *, **Sorina Maciuca**[3], **Zamin Iqbal**[1] *

**1** EMBL-EBI, Hinxton, United Kingdom, **2** Laboratory of Biology and Modelling of the Cell, CNRS UMR 5239, Ecole Normale Supérieure de Lyon, Lyon, France, **3** Genomics England, London, United Kingdom

* brice.letcher@ens-lyon.fr (BL); zi@ebi.ac.uk (ZI)

**Data Availability Statement:** The authors confirm that all data underlying the findings are fully available without restriction. All code and input data tables are publicly-available on github at https://github.com/iqbal-lab-org/paper_pfalciparum_DBs,

## Abstract

While the malaria parasite *Plasmodium falciparum* has low average genome-wide diversity levels, likely due to its recent introduction from a gorilla-infecting ancestor (approximately 10,000 to 50,000 years ago), some genes display extremely high diversity levels. In particular, certain proteins expressed on the surface of human red blood cell–infecting merozoites (merozoite surface proteins (MSPs)) possess exactly 2 deeply diverged lineages that have seemingly not recombined. While of considerable interest, the evolutionary origin of this phenomenon remains unknown. In this study, we analysed the genetic diversity of 2 of the most variable MSPs, DBLMSP and DBLMSP2, which are paralogs (descended from an ancestral duplication). Despite thousands of available Illumina WGS datasets from malaria-endemic countries, diversity in these genes has been hard to characterise as reads containing highly diverged alleles completely fail to align to the reference genome. To solve this, we developed a pipeline leveraging genome graphs, enabling us to genotype them at high accuracy and completeness. Using our newly-resolved sequences, we found that both genes exhibit 2 deeply diverged lineages in a specific protein domain (DBL) and that one of the 2 lineages is shared across the genes. We identified clear evidence of nonallelic gene conversion between the 2 genes as the likely mechanism behind sharing, leading us to propose that gene conversion between diverged paralogs, and not recombination suppression, can generate this surprising genealogy; a model that is furthermore consistent with high diversity levels in these 2 genes despite the strong historical *P. falciparum* transmission bottleneck.

## Introduction

*Plasmodium falciparum* is a single-celled eukaryotic parasite causing malaria disease in humans. Malaria burden remains high worldwide, with 241 million cases and 627,000 deaths in 2020 according to WHO [1]. The high burden is in part due to *P. falciparum*'s ability to evade the human immune system, mediated by 2 main mechanisms [2]. Firstly, cell-surface–exposed antigens targeted by the immune system are produced by functionally redundant gene families. For example, merozoites, the parasite life stage infecting human red blood cells

and frozen on zenodo at https://zenodo.org/doi/10.5281/zenodo.7677547. The github repository implements our genotyping pipeline and our analysis of DBLMSP and DBLMSP2 sequences, and contains Snakemake workflows to reproduce all steps, including downloading the input data using input tsv tables. The data underlying all main and supplementary Figures is also available on zenodo. On github, the input tsv files are located under 'analysis/input_data/sample_lists' (including ENA run accessions; see repository 'README.md' file for details), and the main ones are also copied on zenodo. The set of all DBLMSP and DBLMSP2 sequences we analysed are available on zenodo in the file named 'output_analysed_sequences.tar.gz'. All MosaicAligner images are also available on the github repository and on zenodo as the file named 'figures_recombination_breakpoints_all.pdf'. All software used and versions are stored in the 'reproducibility/container' folder of the github repository, including a definition file for building a Singularity image used by all Snakemake workflows. A copy of this image is also available on zenodo as the file named 'singu.sif'.

**Funding:** BL was funded by a predoctoral fellowship from the European Molecular Biology Laboratory. ZI was funded by a Wellcome Trust/Royal Society Sir Henry Dale Fellowship, grant number 102541/A/13/Z. The funders had no role in study design, data collection and analysis, decision to publish, or preparation of the manuscript.

**Competing interests:** The authors have declared that no competing interests exist.

**Abbreviations:** CNV, copy-number variation; DSR, DBL-spanning region; EBA, erythrocyte binding antigen; ML, maximum-likelihood; MOI, multiplicity of infection; MSP, merozoite surface protein; RBC, red blood cell.

(RBCs), use different members of the Rh and EBA families for invasion [2], and different members of the *var*, *rifin*, and *stevor* families enable infected RBCs to bind to the host micro-vasculature [3]. Secondly, surface antigens are highly diverse at the sequence and immunological levels. In the *var* family, diversity is mainly generated by frequent recombination and gene conversion (sequence copy-pasting) events, occurring both between orthologs during sexual reproduction, and paralogs on the same genome during asexual replication [4–7].

Historically, several cell-surface antigens called merozoite surface proteins (MSPs) were found to display unusual genealogies, with exactly 2 deeply diverged lineages: This includes MSP1, MSP2, MSP3, and MSP6 [8–11]. Such deep divergence suggests ancient origins and possible maintenance by balancing selection for immune escape [12,13], but Roy and colleagues showed that neither this nor neutral evolution should produce exactly 2 lineages, and with such a deep most recent common ancestor [14]. In addition, loci with such extreme diversity levels are at odds with *P. falciparum*'s overall low diversity levels, likely due to its very recent origin (10,000 to 50,000 years ago) in humans from a common ancestor with gorilla-infecting *P. praefalciparum* [15–17].

In this study, we focussed on 2 MSPs called DBLMSP and DBLMSP2, both among the most diverse genes in *P. falciparum* [18], and both encoding cell-surface–exposed antigens recognised by the human immune system [19,20]. They are part of an 8-gene tandemly arrayed family of paralogs, as identified from sequence sharing: All 8 genes possess an N-terminal signal sequence, 6 (including DBLMSP and DBLMSP2) possess a C-terminal SPAM domain, and DBLMSP and DBLMSP2 further uniquely possess a DBL domain [20] (illustrated in S1 Fig). DBL domains mediate a number of important malarial host–pathogen interactions, including between erythrocyte binding antigen (EBA) gene products and RBCs during invasion [21,20], and between *var* gene products on infected RBCs and various human receptors, enabling sequestration [22]. However, their function in DBLMSP and DBLMSP2 remains largely unknown [23].

The evolutionary history of *P. falciparum* surface antigens, including DBLMSP and DBLMSP2, has been difficult to study until now because of reference bias: Reads spanning highly diverged nonreference alleles fail to align to a reference genome, making them hard to reconstruct. To address this, we previously developed gramtools, a software for mapping reads and genotyping using a genome graph incorporating multiple references simultaneously [24,25]. In this study, we developed a new pipeline combining local assembly to reconstruct DBLMSP and DBLMSP2 alleles together with gramtools for comprehensive genotyping. Applying it to Illumina population sequencing data, we assembled the first comprehensive set of alleles for these genes, across >3,500 global *P. falciparum* samples. Studying these in detail, we found that although DBLMSP and DBLMSP2 have diverged substantially, 1 specific region (the DBL domain) contains sequences shared across both genes. We found clear evidence this was driven by gene conversion of DBL sequence between the 2 genes, thus creating highly diverse gene lineages despite the recent gorilla-to-human transmission bottleneck. Interestingly, we also found evidence that DBLMSP2 may have evolved a constrained function specifically in humans.

For the remainder of this paper, we refer to DBLMSP and DBLMSP2 collectively as DBLMSP1/2.

## Results

### 1. New genotyping pipeline outperforms state of the art

To analyse variation in DBLMSP1/2, we used data from malariaGEN, a consortium releasing Illumina whole-genome sequencing data from global *P. falciparum* samples [26]. We used the 2021 data release, consisting of >7,000 samples [27]; of these, we retained 3,589 samples

passing MalariaGEN's quality controls and inferred as being clonal, as multiple infections are common in *P. falciparum* and can confound genotyping [26,28]. These samples come from a total of 29 countries (S2 Fig). After read preprocessing (see Methods), all 3,589 samples were processed using a newly developed pipeline. This pipeline uses our existing genome graph-based tool, gramtools, which we previously showed is effective for genotyping highly diverse genes [25,29], together with assembly-based tools to reconstitute diverged nonreference alleles (see S4 Fig and associated text for details).

To evaluate our genotype calls, we implemented 2 orthogonal approaches (see Methods) and compared our pipeline outputs with those from malariaGEN's existing pipeline, based on GATK. GATK is a state-of-the-art genotyping framework [30,31] but can suffer from reference bias, notably in *P. falciparum* [18]. We found our pipeline clearly outperformed GATK on our target DBLMSP1/2 genes (S6–S8 Figs). Using stringent criteria, we derived a set of DBLMSP1/2 sequences that we deemed "confidently resolved" (see Methods for details; sequences available with this paper, see Data availability). For the GATK-based pipeline, 49% (DBLMSP) and 12% (DBLMSP2) of all sequences were confidently resolved, while for our new pipeline, >81% were confidently resolved for both genes (S9 Fig). Our new sequence set also contained much more variation, as we show in the next section.

## 2. The DBL domain of DBLMSP1/2 is highly variable and contains shared and private sequences

To analyse polymorphism levels in DBLMSP1/2, we translated all confidently resolved gene sequences from the 2 pipelines into proteins and computed 2 measures of sequence diversity, as shown in Fig 1 (measures computed from multiple-sequence alignments). In panel (a), we show within-gene heterozygosity (y-axis), defined as the probability that, for a given gene and at a given aligned position (x-axis), 2 randomly chosen amino acids from the population differ. For both DBLMSP and DBLMSP2, much less diversity was recovered by the GATK-based pipeline (left-hand side panels) compared to ours (right-hand side). In our sequences only, a central region of each gene is particularly polymorphic and spans their DBL domain, delimited with blue vertical dotted lines (see Methods for annotation).

In panel (b), we show between-gene heterozygosity, defined as the probability that, for 2 genes at an aligned position, 2 randomly chosen amino acids—one from each gene—differ. A value of 1 indicates no amino acids in common between the sequences of the 2 genes (fully diverged position), while a value of 0 indicates a single identical amino acid is found in both genes (fixed position). While many positions are fully diverged, a region spanning the DBL domain, shown with red vertical dotted lines, has zero fixed differences between the genes, indicating sequence sharing. This observation is impossible with previous methods (i.e., the GATK-based or any single-reference/non-pangenomic pipeline). We call this region the DBL-spanning region, or DSR, and focus the rest of the analysis in this paper on this region, and using our gramtools-pipeline results.

We then built a hierarchical clustering tree from all unique protein sequences (278 in total) in the multiple-sequence alignment of DBLMSP1/2 to visualise the sequence relationships in the DSR (Fig 2). The tree clearly shows 3 main lineages, marked A, B, and C. Lineages A and C consist exclusively of DBLMSP (yellow in innermost coloured ring) and DBLMSP2 (blue) sequences, respectively, while lineage B contains sequences from both genes. We thus call lineage B "shared lineage" and lineages A and C "private lineages" (for details, see S11 Fig). We note that the shared lineage is abundant in populations: It occurs at approximately 25% to 50% frequencies in all 16 countries with more than 50 samples (S12 Fig). This is consistent with balancing selection maintaining both shared and private lineages in populations.

**a)** Within-gene heterozygosities

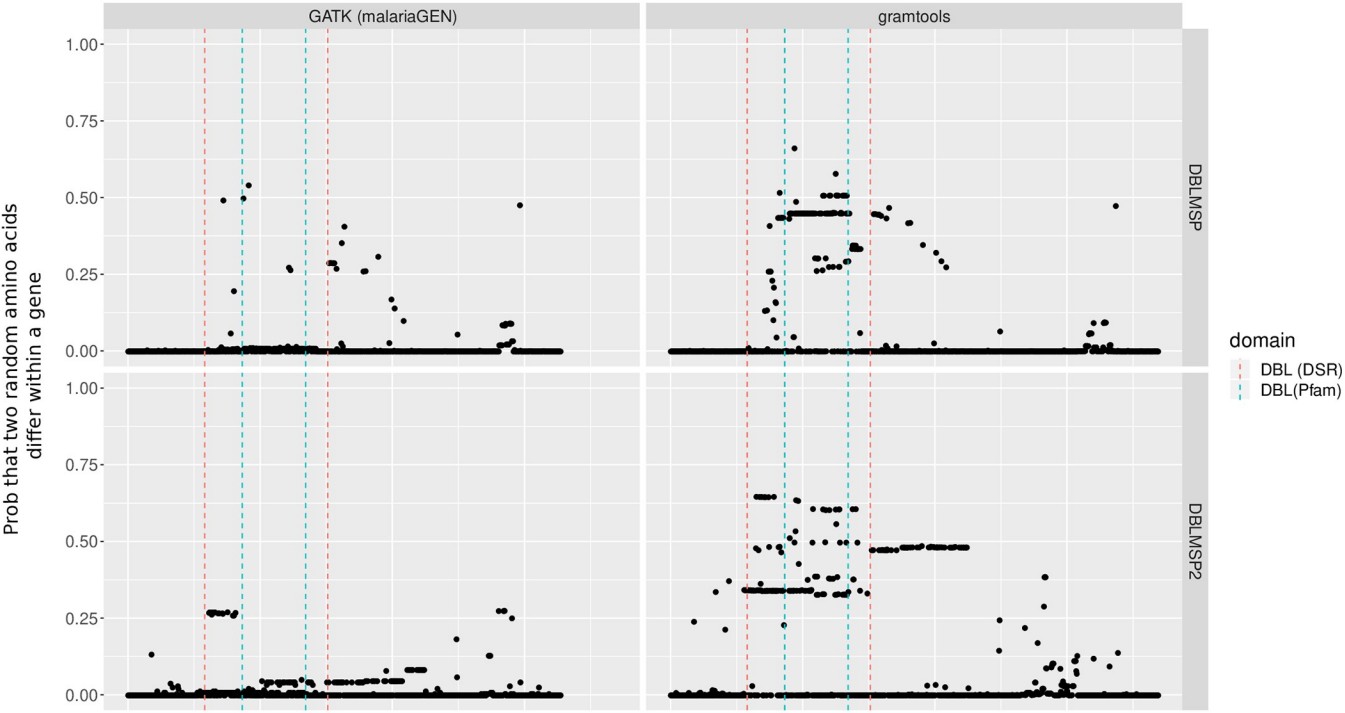

**b)** Between-gene heterozygosities

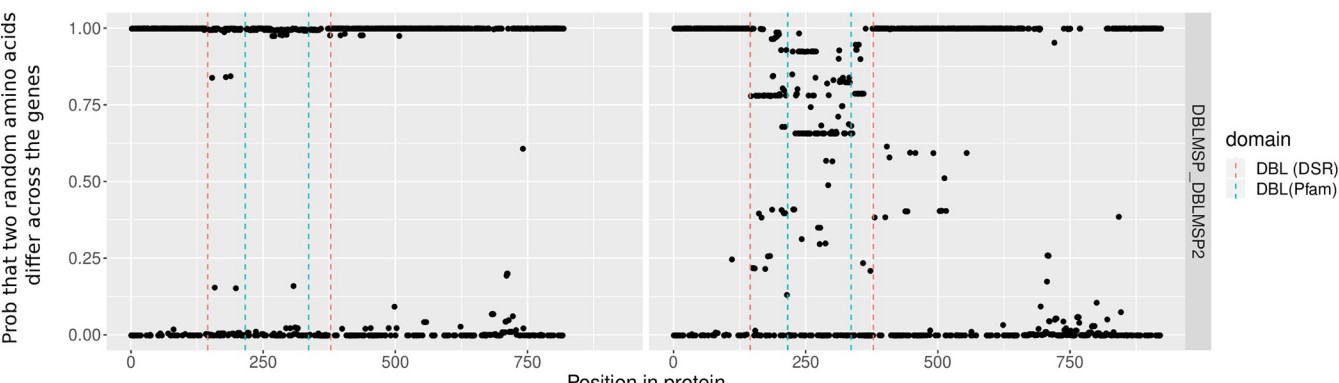

**Fig 1. Within- and between-gene heterozygosities in DBLMSP and DBLMSP2.** In panel (**a**), the y-axis measures the probability that, at each aligned protein position (x-axis), 2 randomly chosen amino acids differ, for each gene and each of the GATK- and gramtools-based pipelines. A region of extreme diversity spans the DBL domain, annotated with blue vertical dotted lines, and is only visible with our new pipeline. Panel (**b**) shows the probability that 2 randomly chosen amino acids, one from each gene, differ. A value of 1 indicates no amino acids in common, i.e., full divergence of the 2 genes. The DBL domain lies in a region of shared sequence, where no amino acid has fully diverged, and indicated with red vertical dotted lines—we call this the DBL-spanning region (DSR). We note that a smaller C-terminal region also displays positions with putative sequence sharing, but these are in fact gap characters in an indel-rich region of the alignment. The data and code to generate this Figure can be found at https://zenodo.org/doi/10.5281/zenodo.7677547.

To probe the evolution of DBLMSP1/2, we searched for orthologs in the 6 closest known relatives of *P. falciparum* (all part of subgenus *Laverania*). Using sequencing data and genome assemblies from Otto and colleagues [17], we could reconstitute up to 2 sequences per species (see Methods). For DBLMSP, we found clear orthologs in the 3 most closely related species to *P. falciparum*: *P. praefalciparum*, *P. reichenowi*, and *P. billcollinsi* (in increasing order of divergence time from *P. falciparum*), but not in the more diverged species. For DBLMSP2, we

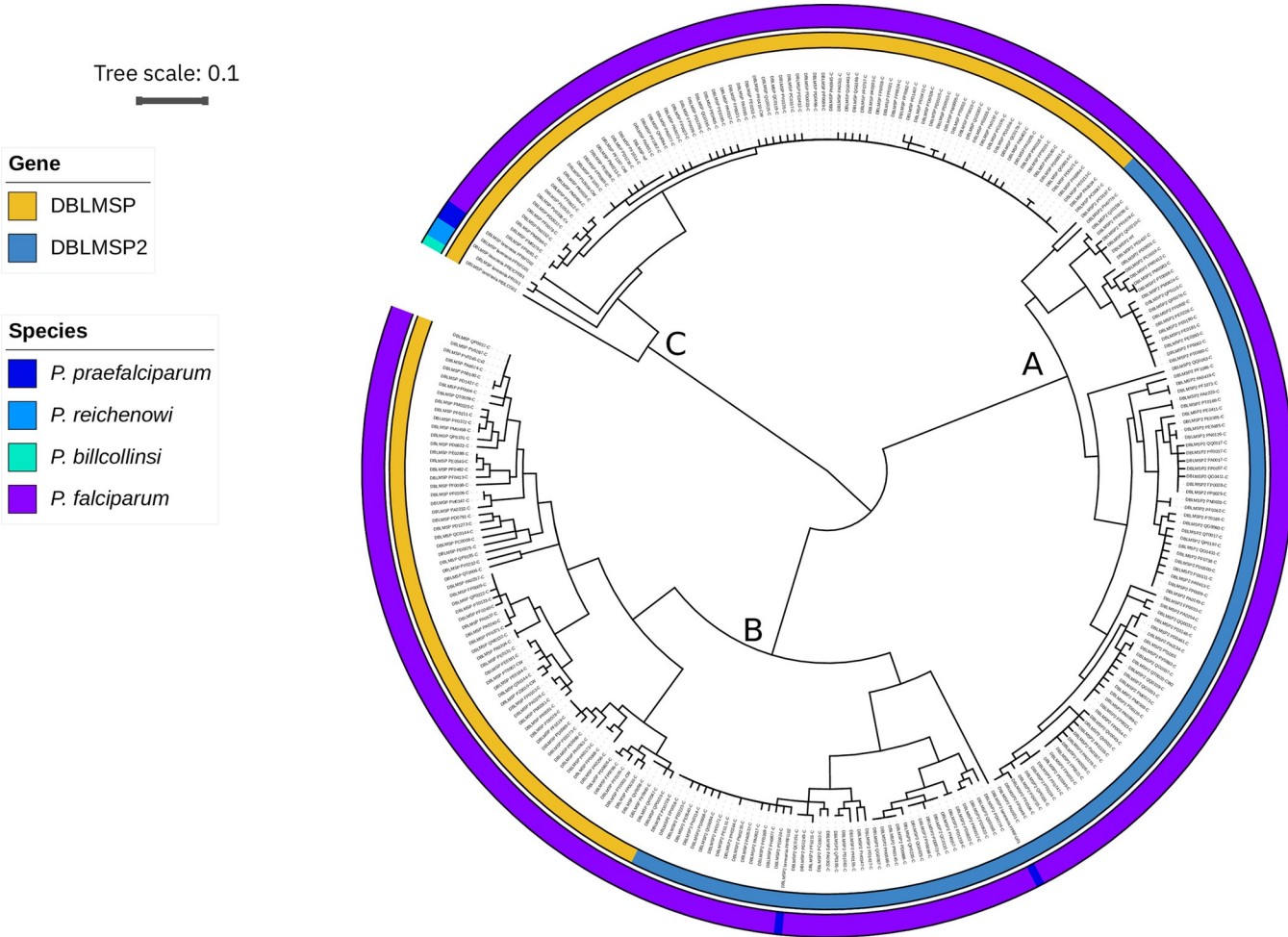

**Fig 2. Deeply diverged private and shared lineages in DBLMSP1/2.** We built a hierarchical clustering tree of all unique DBL-spanning protein sequences (see Methods). The inner ring colours sequences by gene of origin (DBLMSP, DBLMSP2), and the outer ring shows species of origin, for *P. falciparum* and its 3 most closely related species. Three main lineages exist in the tree, labelled A, B, and C: Lineages A and C contain only representatives of DBLMSP2 and DBLMSP, respectively ("private lineages"), and lineage B contains representatives of both ("shared lineage"). The data and code to generate this Figure can be found at https://zenodo.org/doi/10.5281/zenodo.7677547.

found a clear ortholog in *P. praefalciparum* and *P. reichenowi* only. We thus date the ancestral duplication of DBLMSP to DBLMSP2 to between the *P. falciparum*–*P. billcollinsi* and *P. falciparum*–*P. reichenowi* splits and thus before the jump of *P. falciparum* from gorilla to human.

The species of origin of each DBLMSP/DBLMSP2 sequence is shown in Fig 2 (outer ring). For DBLMSP, the orthologs fall in a distinct sublineage of lineage C and consistently with the known *Laverania* phylogeny. For DBLMSP2, the 2 orthologs in *P. reichenowi* are pseudogenes (consistent with prior knowledge [17]) and not shown in the tree (premature stop codons near the start codon make the DSR incomparable). The 2 identified orthologs in *P. praefalciparum* both fall in the shared lineage B: One has a premature stop codon (rightmost *P. praefalciparum* sample in lineage B), and the other lies nested inside a clade of *P. falciparum* alleles. We found this is consistent with *P. falciparum*'s recent origin from *P. praefalciparum* via a tight bottleneck: in trees of full-length DBLMSP and DBLMSP2 sequences and of a single-copy, well-conserved gene (AMA1), *P. praefalciparum* alleles also nest inside *P. falciparum* clades (in contrast to alleles from more diverged species; S14 and S15 Figs). Our data also suggest that DBLMSP2 may have evolved a conserved function in *P. falciparum* only; we return to this in the discussion.

Overall, in *P. falciparum*, 2 deeply diverged lineages exist per gene, one of which is shared across both—leading to 3 lineages in total, instead of 4. In S16 Fig, we show HMM logos of what each sequence lineage prototypically looks like at the amino acid level. In other highly diverse MSPs, recombination between the 2 lineages has been reported to be rare or absent [11,32]; we next formally test for recombination in DBLMSP1/2, as well as gene conversion as a putative driver of the shared lineage.

## 3. Recombination and gene conversion in the DBL domain

To detect recombination in our DBLMSP1/2 protein sequences, we used a method developed by Zilversmit and colleagues [33] for studying the *var* genes in *P. falciparum*. Briefly, each sequence in a panel is aligned to all others using an HMM-based model that performs pairwise alignment between target and donor, while allowing for switching between donors (i.e., recombination). Given our high number of sequences, we first clustered them into 35 representatives (at 96% identity), as sequences that are too closely identical would only get aligned to each other, obscuring more distant recombination. We ran Zilversmit and colleagues' implementation MosaicAligner on this panel and built visual representations of the outputs to verify each inferred breakpoint (code available with this paper; see Data availability). In Fig 3A, we show one such "mosaic alignment" in detail. The target, a DBLMSP sequence (second row), is a recombinant of 2 other DBLMSP donors, and the vertical red line shows the inferred recombination breakpoint. Either side of the breakpoint, the highlighted donor has fewer mismatches (red-coloured letters) than the nonhighlighted one; this held for all 35 alignments (S18 Fig). In panel (b), we show 3 full mosaic alignments: one for the first representative of each gene, and one for which the breakpoints span donors on both genes (discussed next). Illustrations for all 35 mosaic alignments are available with this paper (see Data availability). Overall, across the 35 alignments, we found breakpoints at a total of 13 (DBLMSP) and 15 (DBLMSP2) of 254 positions in the DSR, with no apparent hotspot structure as they were spread across the full region (panel c). While most recombination events occurred between orthologs and inside the 3 main lineages of Fig 2, we also observed a few recombination events between orthologs in different lineages (A-B and C-B; S19 and S20 Figs).

For 3 of the mosaic alignments, the donor sequences came from different genes (last alignment of Figs 3B and S21), consistent with sequence exchange between the paralogs. This can notably occur during repair of double-strand breaks and subsequent sequence pasting—also called gene conversion—from a nearby unbroken template. The template is usually a homologous gene copy, either from an identical sister chromatid (e.g., after genome replication) or from a homologous chromosome (e.g., during recombination in meiosis). In some cases, a nearby paralog can act as the template instead; this is also called nonallelic conversion [34]. We performed a test to detect nonallelic gene conversion between DBLMSP and DBLMSP2. For each of the 2,882 samples in which both DBLMSP1/2 genes were confidently resolved, we pairwise aligned the 2 genes and measured sequence identity at the DNA codon level (see Methods). In Fig 4B, we illustrate the 209 samples for which the fraction of identical codons in the DSR was high (>0.5; see S22 Fig for full distribution), after ruling out duplications of the DSR having occurred inside a gene (see Methods). Each row shows 1 sample's DBLMSP1/2 sequence alignment in the DSR, and each column shows 1 codon, with cells coloured beige for identical codons and black for different codons (illustrated in panel a). Stretches of near-uninterrupted beige are clearly visible, supporting gene conversion between DBLMSP1/2.

The samples in Fig 4 panel b are consistent with 2 main conversion events having occurred ancestrally, with different breakpoints (start and end positions of the beige strips) and sequences (S23 Fig). Samples from the 2 events are geographically widespread, occurring

**a)** Example of mosaic alignment with recombination

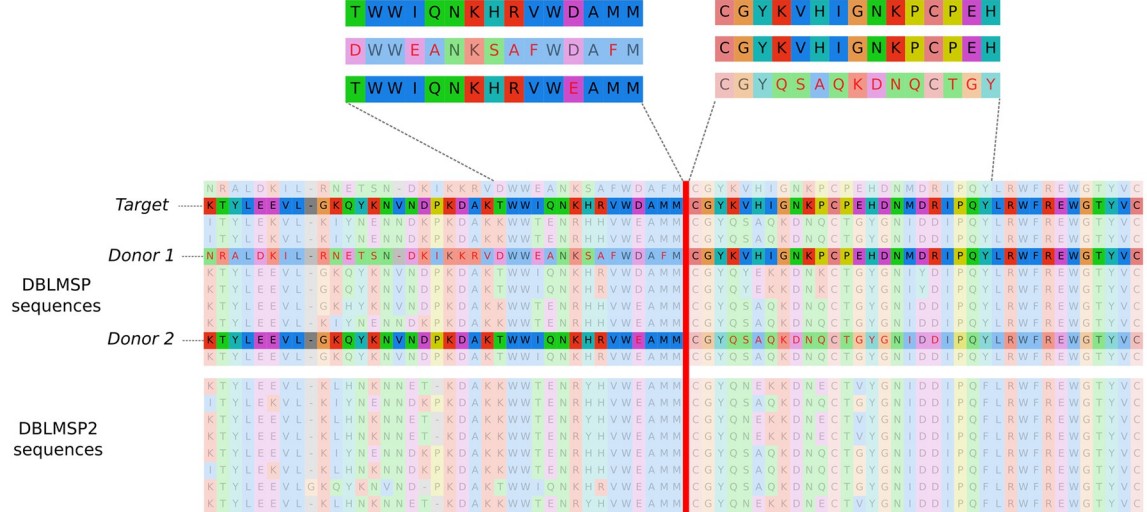

**b)** Mosaic alignments for 3 of the 35 representatives

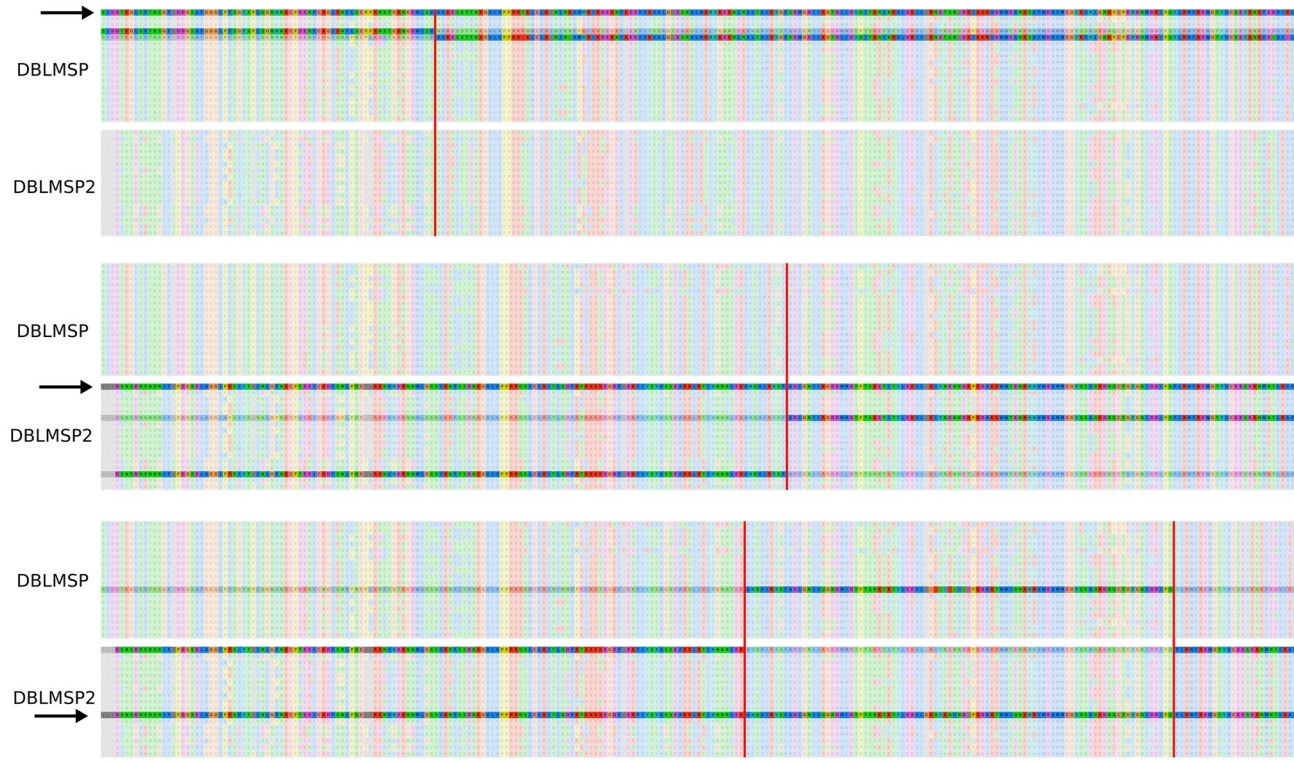

**c)** Aggregated breakpoints

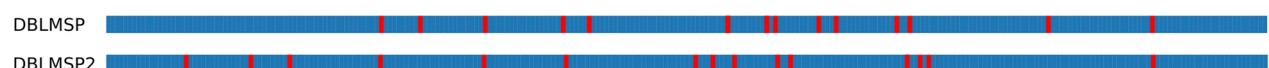

**Fig 3. Mosaic alignments reveal widespread recombination in DBLMSP1/2.** (**a**) Visual confirmation of MosaicAligner's inferred alignment for the first DBLMSP sequence (*target*; fully shaded sequence), aligned to 2 other DBLMSP sequences (*donors*; partially shaded sequences). The red vertical line marks the switch in alignment between the 2 donors. Either side of the switch, the target aligns to the shaded donor with many fewer edits than to the other donor (red-coloured letters flag mismatches). (**b**) Full mosaic alignments illustrated for 3 out of 35 representative sequences in the panel. In each panel, the aligned target is the fully opaque row labelled with an arrow, and the donors are shown as partially opaque rows. Illustrations for all 35 alignments are

available with this paper. (**c**) For each gene, the aggregated locations of all breakpoints from the mosaic alignments are shown. The breakpoints did not seem to cluster into hotspots. The data and code to generate this Figure can be found at https://zenodo.org/doi/10.5281/zenodo.7677547.

across both west and east sub-Saharan Africa and Southeast Asia (S24 Fig), suggesting they are both being actively maintained, either through selection or recurrent conversion.

### 4. Diversification of DBLMSP1/2 through gene conversion

In Fig 5, we illustrate the relationship between gene conversion and the lineages of DBLMSP1/2. In panel a, we show once more the clustering tree from Fig 2, with the addition of an outer ring marking samples belonging to the 2 different conversion events identified in Fig 4. The coloured stars mark the putative locations of new subclades in the tree that were created by gene conversion.

Conversion event 1 is labelled in green in Fig 5A, and panel b illustrates its effect on an ancestral tree of DBLMSP1/2 sequences, depending on whether DBLMSP pasted into DBLMSP2 (scenario i), or vice versa (scenario ii). For example, in scenario ii, a preexisting sequence from lineage B.2 in DBLMSP2 pasted into a sequence from DBLMSP lineage C, giving rise to lineage B.1 in DBLMSP and the creation of 2 deeply diverged lineages in DBLMSP. In scenario i, 2 deeply diverged lineages are instead created in DBLMSP2, through pasting from a DBLMSP allele in lineage B.1. Note that because almost all of the sequence has been pasted (approximately 80%; Fig 4B), the recipient sequence ends up in a lineage close to the donor sequence; the opposite would hold if a small minority (e.g., 20%) of the sequence had been pasted. We hypothesise scenario ii is more likely, leading to the birth of subclade B.1 (green star in Fig 5A), because we identified sequences from *P. praefalciparum* in these 2 lineages (B.2 and C), but no sequences from either lineages B.1 (DBLMSP) or A (DBLMSP2).

Similarly, conversion event 2 led to the birth of lineage A.1 or lineage B.1.1 and likely occurred after conversion event 1 as subclade B.1.1 lies nested within subclade B.1. Here, we cannot speculate on which gave rise to the other, so we show 2 pink stars in Fig 5 panel a. Note that because of the intermediate fraction of pasted sequence for this event, (approximately 0.55; Fig 4B) subclades A.1 and B.1.1 are not located close to each other in the tree. Finally, while our data are clearly consistent with sublineage birth in the tree of DBLMSP by gene conversion, they do not explain the preexistence of deeply diverged lineages in DBLMSP2 (Fig 5B). We return to this in the discussion.

### 5. Testing for direct evidence of evolution in DBLMSP1/2

The recombination and gene conversion events studied so far were inferred indirectly from population-level data. To test for direct evidence of these events in DBLMSP1/2, we also analysed data from repeatedly sequenced isolates through time. We looked for mutations in DBLMSP1/2 in 2 sources: the "clone trees" from Hamilton and colleagues [35], who repeatedly cloned, cultured and sequenced individual isolates (spanning approximately 700 erythrocytic life cycles in total), and 4 experimental genetic crosses between different strains of *P. falciparum* (142 sequenced parents and progeny in total [18,36]; see Data availability and Methods for details). Overall, we found just 2 point mutations in one of the genetic cross progeny and no direct evidence of recombination or gene conversion in DBLMSP1/2 (S27 Fig). We note, though, that gene conversion in genes RH2a and RH2b was observed in a repeatedly cultured and sequenced isolate by Cortes [37].

### Discussion

The existence of exactly 2 deeply diverged lineages that have not recombined in specific *P. falciparum* genes, historically called "allelic dimorphism" in the malaria literature, has been a

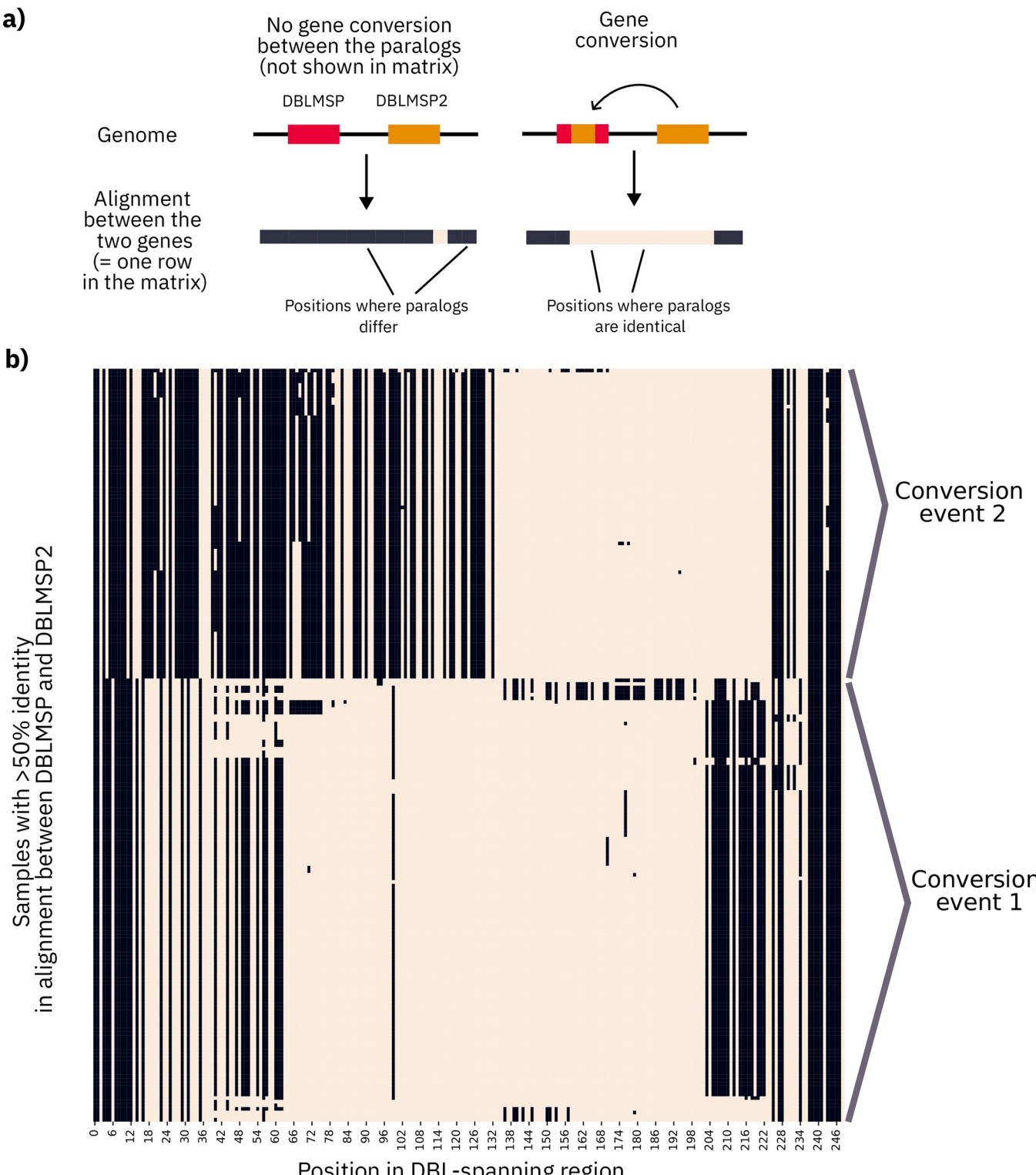

**Fig 4. Evidence for nonallelic gene conversion between DBLMSP and DBLMSP2 in the DSR.** (**a**) This scheme explains the matrix that follows in panel (b). For each of the samples in which both DBLMSP1/2 gene sequences were confidently resolved, we aligned their DNA sequences in the DSR and recorded positions where codons were identical between the 2 genes (beige cells) versus different (black cells). Gene conversion should appear as contiguous strips of beige cells. (**b**) The 209 samples with >50% identical codons between DBLMSP and DBLMSP2 are shown (rows) at each position of the DSR (columns). The strips of near-all beige indicate likely sequence copying between the 2 genes in a sample, supporting within-genome gene conversion. Two main sets of samples

can be distinguished visually, consistent with at least 2 distinct conversion events (labelled on the right) having occurred in the lineages leading to these samples. The data and code to generate this Figure can be found at https://zenodo.org/doi/10.5281/zenodo.7677547.

long-standing puzzle [8,14]. While balancing selection can maintain diverged lineages via host immune pressure, intermediate forms are expected through recombination if it is not suppressed, and more than 2 lineages should occasionally be observed. Here, we showed that DBLMSP and DBLMSP2 each display 2 deeply diverged lineages (Fig 2) in *P. falciparum*, with each lineage maintained at high frequencies across populations, consistent with balancing selection. With the benefit of our large global dataset and fully resolved alleles, we also found extensive polymorphisms within each lineage, i.e., more than strictly 2 lineages per gene, as well as extensive recombination, mainly within each lineage, but also between diverged lineages of the same gene. Rather than a process of divergence without recombination, we propose that allelic dimorphism can instead be generated rather abruptly by gene conversion between diverged paralogs.

This idea is consistent with the recent introduction of *P. falciparum* into humans by zoonosis from a common ancestor with the gorilla-infecting *P. praefalciparum*, only about 10,000 to 50,000 years ago [15]. *P. falciparum* indeed has very low genome-wide diversity levels (approximately 10-fold lower than other species of the subgenus *Laverania* [17,38]), possibly through the jump of only one or a few individuals into humans [17,16,39]. It is thus possible that the DBLMSP and DBLMSP2 sequences we reconstituted from *P. praefalciparum*, which fall in exactly 2 lineages, represent the ancestral lineages of *P. falciparum*. We found that at least 2 new sublineages were generated by paralogous gene conversion (Fig 5). While we cannot definitively infer the direction of conversion, for 1 sublineage, it is likely that DBLMSP2 pasted into DBLMSP. A subsequent conversion event of unknown direction gave rise to further lineage diversity. Our data do not resolve the preexisting deep split between lineages A and B in DBLMSP2, however. One possibility is that another DBL domain-containing gene in *P. falciparum* gave rise to lineage A, though testing this will likely require long-read data, especially to resolve the *var* gene DBL domains.

In the future, our evolutionary model could also be tested in other MSPs that have been called "dimorphic": notably, MSP2 occurs in tandem with another MSP (MSP4), and MSP3 and MSP6 both occur in the same 8-gene paralog tandem array as DBLMSP and DBLMSP2 (S1 Fig). Interestingly, and as for DBLMSP1/2 here, in 2003, Nielsen and colleagues reported gene conversion between the paralogous genes FP2A and FP2B located 10 kbp apart, causing the genes to look far more diverse than consistent with a recent bottleneck [40]. We do not propose that our model is exhaustive, however: other genes with deeply diverged lineages, like MSP1 or EBA-175, do not occur in tandem with a paralog [41,42]. More generally, our model does not imply that exactly 2 lineages should exist, nor that they do not recombine, both of which would be hard to expect in the long term.

In terms of evolutionary constraints, we note that in *P. falciparum*, DBLMSP2 appears more highly constrained than DBLMSP: of 234 DBLMSP1/2 gene sequences with premature stop codons, 196 lie in DBLMSP and 38 in DBLMSP2. By contrast, in *P. reichenowi* and *P. praefalciparum*, all 4 identified DBLMSP orthologs had a complete open reading frame (2 in each species), while 1 of 4 DBLMSP2 orthologs did (in *P. praefalciparum*). This raises the intriguing possibility of a human-specific function (or constraint) having evolved in DBLMSP2. To fully test this will require population-level data in *P. reichenowi* and *P. praefalciparum*.

We also note that gene conversion has only occurred (or been selected in) the DSR, while the rest of the DBLMSP1/2 genes have diverged substantially (Fig 1). In other *P. falciparum*

a) Conversion events in the clustering tree built on DSR of DBLMSP1/2

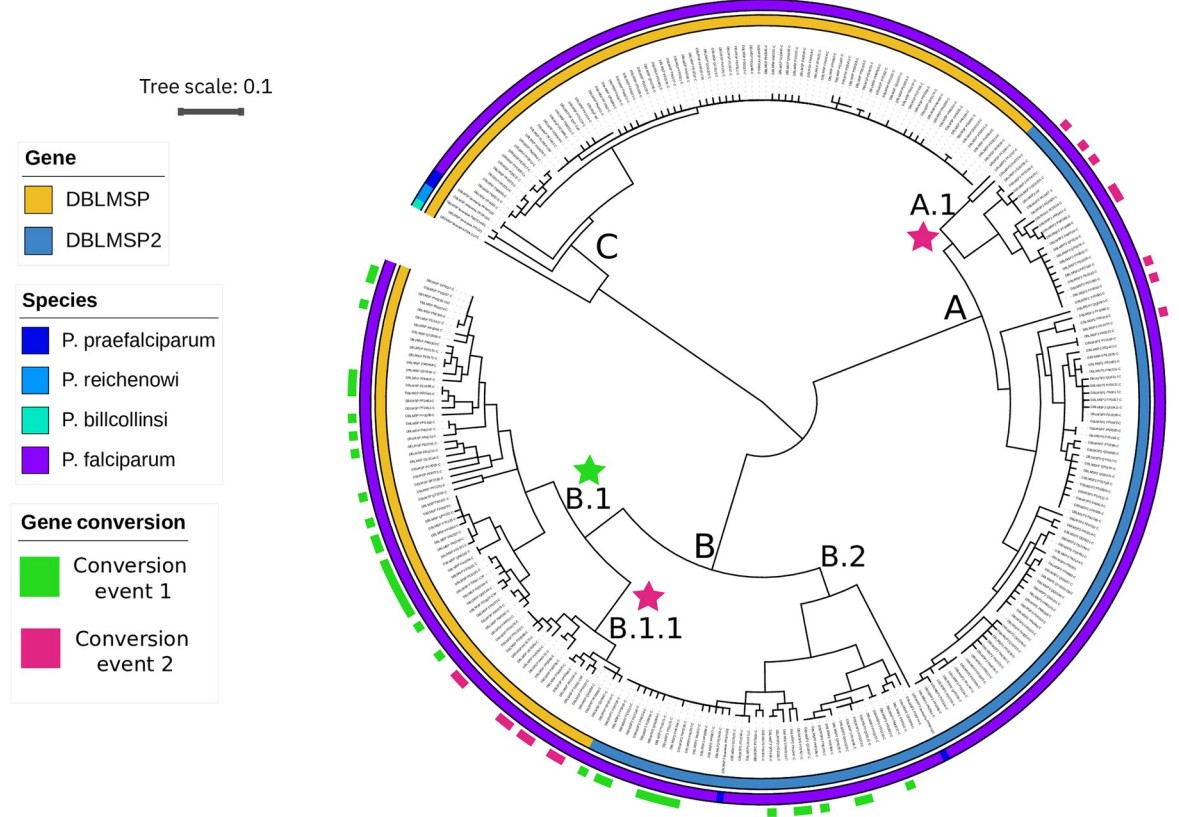

b) **Paralogous gene conversion creates new sub-clades**
Example in conversion event 1: creation of sub-clade B.2 **or** B.1

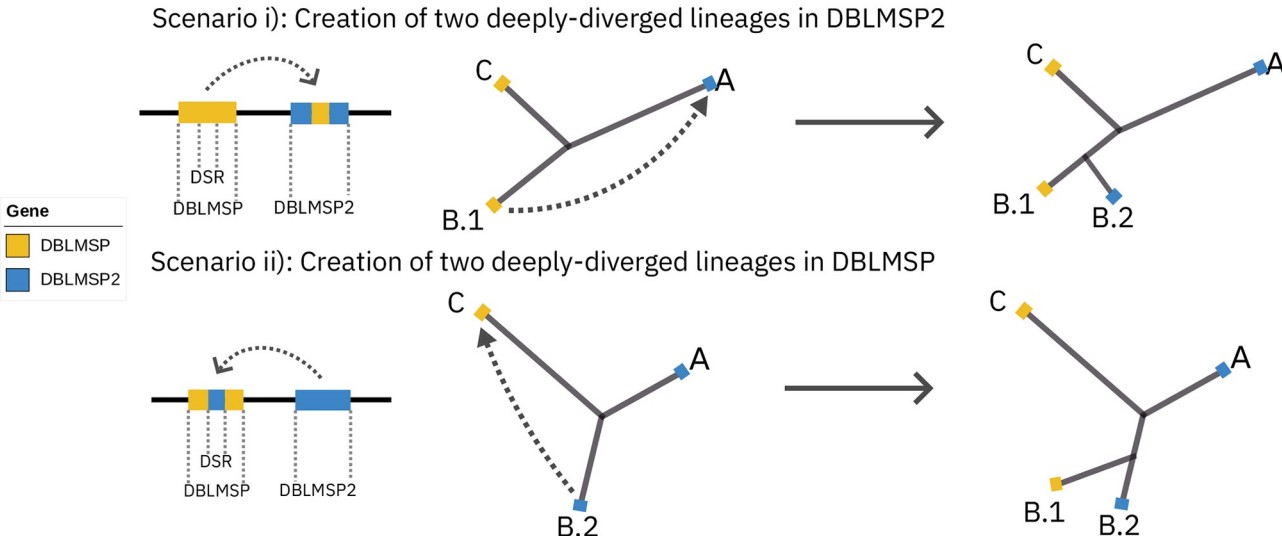

**Fig 5. Sublineage birth by gene conversion between DBLMSP and DBLMSP2.** (**a**) The same clustering tree as in Fig 2 is shown (built from alleles of the DSR), with the addition of an outer ring labelling the sequences shown in Fig 4. These were divided into the 2 distinct conversion events identified in Fig 4 and labelled green (conversion event 1) and pink (conversion event 2). The 2 events gave rise to new subclades in the tree; for how, see the main text and the following panel. (**b**) A simplified schematic showing how gene conversion event 1 created 2 deeply diverged lineages either in DBLMSP2 (scenario i) or in DBLMSP (scenario ii), depending on the direction of the sequence pasting. The data and code to generate this Figure can be found at https://zenodo.org/doi/10.5281/zenodo.7677547.

proteins, the DBL domain is key to parasite invasion and persistence: In the EBA family, the DBL domain mediates RBC invasion, and in the *vars*, it binds to receptors on endothelial cells and other infected RBCs (iRBC), enabling iRBC sequestration [3,21]. Its function in DBLMSP and DBLMSP2 is not known, although one study found it binds to human IgM [23]. As an alternative to selection for immune evasion, it could be that the shared DBL domain lineage mediates binding to a shared environment, in which the 2 genes are coexpressed, while the private lineages mediate binding to private environments. This could be consistent with DBLMSP being expressed in blood-stage, asexual merozoites, and DBLMSP2 being likely only expressed in a subset of schizonts committed to gametocytogenesis (sexual cycle; [43]); gametocytes are reported to preferentially occur in specific niches like the human bone marrow [44]. This could perhaps be tested biologically (e.g., using AVEXIS [45,46]), or computationally, using future protein–protein interaction prediction methods based on AlphaFold, for example [47].

In conclusion, our study highlights the importance of paralogous gene evolution, and we hope that the higher-resolution sequence data in DBLMSP1/2 will help contextualise their biological function once it is elucidated.

## Methods

### For results section 1

**Sample preprocessing.**   Of the >7,000 samples available in the MalariaGEN release that we analysed (released November 2020 [27]), 5,970 were labelled as analysis-ready by the consortium (e.g., after filtering out samples where less than 50% of the genome was callable due to coverage, or samples with evidence of more than one infecting species [27]).

The sequenced life stage of *P. falciparum* is haploid, so a single haplotype can be expected in each sample. However, samples with multiple co-occurring strains (multiplicity of infection (MOI) > 1) are common in *P. falciparum* and hard to genotype confidently. We thus further filtered out samples with evidence of MOI > 1, using the $F_{WS}$ metric [26], which correlates well with MOI in *P. falciparum* [48]. We used the $F_{WS}$ values computed by MalariaGEN [27] and their threshold of > 0.95 for clonality, leaving 3,589 samples for analysis.

**Read preprocessing and quality control.**   For each sample in the analysis set, reads were downloaded from the ENA, trimmed using trimmomatic [49] to remove adapters and low-quality bases from read ends and subsampled to a maximum of 60-fold expected coverage using rasusa [50]. Trimming enables better genotyping with gramtools (see below), and 60-fold coverage is sufficient for genotyping and avoids excess computation. We then characterised the preprocessed reads. Across all 3,589 samples, read lengths had 2 modes, at 75 and 100 bp, and the estimated per-base sequencing error rate had a single mode of $1 \times 10^{-4}$ (S3 Fig, upper panels).

We also aligned reads to the 3D7 reference genome using bwa-mem [51] and measured the average fold-coverage and estimated sequenced fragment lengths (from the distance between aligned paired-end reads). We measured these in alignments falling inside *P. falciparum* "core genome" as defined by [52] by excluding highly repeated or variable genomic regions like the telomeres and *var* genes. We selected a 240-kbp subset of this core genome, spread evenly across all 14 *P. falciparum* chromosomes. Fold-coverage mostly lay between 25 and 50 and sequenced fragment lengths between 200 and 300 bp (S3 Fig, lower panels). Some samples had a fold-coverage <10 and were dealt with in a subsequent filtering step described below.

**Genotyping evaluation and performance.**   We provide only a brief summary of genotyping evaluation and performance here and refer the reader to S1 Text for full details. We used 2 approaches to evaluate the genotype calls made by both pipelines (S4 and S5 Figs). The first used 14 independent samples with both Illumina and PacBio data, from which high-quality

truth assemblies were built in [52]. Using the Illumina data only, we genotyped these samples with both pipelines and compared the calls to the truth assemblies. The second relied on remapping the sequencing reads of our 3,589 analysed samples to an "induced reference genome," made by applying all the calls made by a pipeline to the 3D7 reference genome. We then measure genotyping quality through the level of agreement between reads and the induced genome. From the read pileups in DBLMSP and DBLMSP2 (appropriately translated from 3D7 coordinates) and for each sample, we measured the number of positions with low coverage, the number of positions where the majority-base in the reads differed from that in the induced reference, and the number of reads with large insert sizes (see S1 Text for how we defined large).

We then defined a gene sequence as *confidently resolved* if, across the full gene (DBLMSP: 2,094 base pairs (bp); DBLMSP2: 2,289 bp), the reads contained no positions with less than 5 aligned reads, no positions where the majority of reads disagreed with the induced reference, and <15% of reads with large insert sizes. This resulted in 5,895 confidently resolved DBLMSP1/2 sequences in total for the gramtools-based pipeline (including the two 3D7-reference representatives). We observed that for the gramtools-based pipeline results, a further 200 DBLMSP1/2 sequences (92 DBLMSP and 108 DBLMSP2) had no coverage gaps and <15% reads with large insert sizes, and a single majority-pileup difference. We corrected these single-SNP sequences using a custom script (available with this paper), and added them to our analysis set, and also added the 28 DBLMSP1/2 sequences from the 14 samples assembled by Otto and colleagues [52], giving a total of 6,123 analysed sequences.

Each step in the gramtools-based pipeline gradually improved genotyping performance across both evaluation metrics and confidently resolving most samples at the end (S6 Fig). Overall, the gramtools-based pipeline also clearly outperformed the GATK-based pipeline (S7–S9 Figs).

## For results section 2

**Translation to protein.** DBLMSP1/2 gene sequences were translated to protein using seqkit [53], and any resulting sequence with 2 or more stop codons excluded, as DBLMSP and DBLMSP2 are single-exon genes so that a single stop codon at the end of the sequence is expected (i.e., there is no need to consider stop codons in introns). This removed 3.8% (234/6123; 5,889 analysed sequences left) of the analysis-ready sequences from our gramtools-based pipeline and 8.0% (178/2,223) of those from the GATK-based pipeline. Of our 234 removed sequences, 206 samples had 1 full-length and 1 truncated protein sequence, while in the remaining 14, both protein sequences were predicted to be truncated. We analysed polymorphism levels at the protein level as, proteins being the functional units of cells, they are more likely to be directly under selection.

**Heterozygosities.** The heterozygosities in Fig 1 were computed on multiple-sequence alignments built using mafft [54].

At a given aligned position, we define the set of all observed amino acids as $\{a_1, a_i. . .,a_n\}$, and their frequency in the sequences of gene $j$ as $f_j(a_i)$. Then, the between-gene heterozygosity for genes $j$ and $k$ is $h_{jk} = 1 - \sum_{i=1}^{n} f_j(a_i)f_k(a_i)$.

The within-gene heterozygosity is the equation above evaluated on a single gene $j$ ($h_{jj}$). The latter is classically called simply heterozygosity and is closely related to the "nucleotide diversity," $\pi$ [55].

**DBL domain.** The DBL domain was annotated on the 3D7 sequence of DBLMSP by downloading its HMM model from InterPro (https://www.ebi.ac.uk/interpro/entry/Pfam/PF05424/) and mapping it using hmmscan from the hmmer suite [56].

**Reconstituting sequences from other *Laverania*.**    To obtain DBLMSP and DBLMSP2 sequences from other *Laverania* parasites, we used the data from Otto and colleagues [17], in which 6 *Laverania* species were identified in *Plasmodium* DNA from chimpanzee and gorilla isolates in sanctuaries. For each species, 3 to 4 isolates were short-read sequenced (Illumina), and 1 isolate also long-read sequenced (PacBio) for building chromosome-level assemblies. In total, we could reconstitute 9 DBLMSP1/2 sequences from 3 other *Laverania* species.

To find DBLMSP and DBLMSP2 in the assemblies (plus an additional assembly for *P. reichenowi* [57]; all accessions available with this paper, see Data availability), we mapped the gene sequences from our gramtools-based pipeline to each assembly using minimap2 [58] (preset: "-x asm20"). Using bedtools [59], we merged all overlapping hits into single intervals and extracted the assembly sequences of each merged interval. For the 3 phylogenetically closest species to *P. falciparum* (*P. praefalciparum*, *P. reichenowi*, and *P. billcollinsi*), we could obtain sequences of the same lengths as those in *P. falciparum* (approximately 2 kbp) and starting with a start codon: 1 DBLMSP and 1 DBLMSP2 in *P. praefalciparum*, 2 DBLMSPs and 2 DBLMSP2s in *P. reichenowi*, and 1 DBLMSP in *P. billcollinsi* (7 sequences in total; 5 of which are shown in Fig 2, as the 2 DBLMSP2 sequences from *P. reichenowi* are completely pseudo-genised). For *P.gaboni*, we found a small region (approximately 800 bp) redundantly matching to both DBLMSP and DBLMSP2 that we discarded, and for *P. adleri* and *P. blacklocki*, we obtained no hits. To confirm that DBLMSP1/2 were indeed missing in the more distantly related *Laverania*, we confirmed we could find orthologs of genes in the immediate vicinity of DBLMSP/DBLMSP2 on the *P. falciparum* genome, including MSP11 located in-between DBLMSP and DBLMSP2, as well as AMA1, a single-copy and well-conserved gene (S13 Fig). For AMA1, correct open reading frames were obtained using Liftoff [60].

As *P. praefalciparum* is particularly relevant to the evolution of *P. falciparum*, we worked to resolve DBLMSP and DBLMSP2 in a further 3 isolates with only Illumina data [17]. For each sample, we performed preprocessing as above (trimmomatic + rasusa), genotyped each sample with gramtools on the graph built from the 3,589 "analysis-set" samples above, and then further ran all steps of our gramtools-based pipeline up to and including Gapfiller (S4 Fig). For 1 sample (PPRFG02), both genes were confidently resolved in their DBL domain by our criteria above, and their sequences used in Fig 2. The other 2 samples were not resolved: One had too few reads, likely due to host contamination, and the other was highly mixed with another species (*P. adleri*) (consistent with S1 Table of [17]).

**Clustering trees.**    The hierarchical clustering tree was built using scipy [61], using the Hamming distance (number of differences) between all unique protein sequences in the DSR of the multiple-sequence alignment of DBLMSP1/2 (this is equivalent to the edit distance between unaligned sequences). Sequences were clustered by the "average" method, i.e., iteratively linking the 2 clusters with the smallest average distance. This is also called UPGMA. The tree is thus not a "true" phylogenetic tree—notably, it does not model the probabilities of different mutations or mutation rate variation across sites.

## For results section 3

**Identifying recombination.**    The 5,889 DBLMSP1/2 protein sequences in the DSR were first clustered at 96% identity using cd-hit [62], producing 35 representative sequences. This allows detection of more distant recombination. To see why, consider a sequence A produced by a recombination of 2 highly distinct others, B and C. If another sequence D exists that is one SNP away from A, then A will be aligned full-length to D.

To perform "mosaic alignment," MosaicAligner [33] (originally called Tesserae) uses an HMM model parameterised by amino acid emission probabilities, and transition probabilities

(match/indel transitions and donor switches) that must be estimated. Following the original model specification for MosaicAligner [33], the maximum-likelihood (ML) values for all alignment-related parameters were first estimated with the recombination probability ρ set to zero, and ρ was then estimated as the value for which the sum of all target alignments to the panel was maximal. The ML (Viterbi) path for each of the 35 representatives was then obtained given the inferred ML parameter values. From the textual output from MosaicAligner, visual representations of the alignments (e.g., Fig 3B) were produced using custom code available with this paper.

All mosaic alignments included at least 1 recombination breakpoint. To validate these, we compared the edit distance of each target to its MosaicAligner-inferred donor path with the edit distance to the single closest donor. The former was always smaller than the latter (S18 Fig).

**Identifying gene conversion.** When comparing the sequences of DBLMSP1/2 in individual samples, we aligned their DNA sequences, as gene conversion occurs at the DNA level, and measured the fraction of identical codons, not nucleotides, to match the protein-level analysis. Notably, codon-level identity is closer than nucleotide-level identity to protein-level identity, though a lower-bound of it as 2 identical amino acids can be encoded by 2 different codons.

For identifying samples with evidence of gene conversion, we looked for stretches of identical sequence between the 2 paralogs on the same genome, for all samples where both DBLMSP1/2 sequences were "confidently resolved," meaning (as defined above) no coverage gaps or pileup-based differences and no high levels of large-insert sizes. To rule out the possibility of erroneously attributing sequence sharing to a duplication event in 1 gene, we further filtered out samples with evidence of a possible gene copy-number variation (CNV; see S25 Fig). Of the 212 samples with a codon-level identity >0.5, 3 had a possible CNV, leaving 209 samples all shown in Fig 4.

We further validated 8 samples from each gene conversion event in Fig 4 by manually inspecting read coverage levels and insert sizes in IGV [63]. We found that coverage levels, insert sizes, and read pair orientations were all consistent across DBLMSP, DBLMSP2, and an unrelated gene, AMA1, confirming no duplication of the DBL domain had occurred in these samples.

## For results section 5

For the *P. falciparum* genetic crosses, we used all 4 publically available crosses, between strains 3D7 and HB3 [64], HB3 and Dd2 [65], 7G8 and GB4 [66], and 803 and GB4; the raw data are available in [18] and [36] and listed directly in the Data availability section. For the "clone trees," we used all available data from [7] and [35], building tsv files from their supplementary tables and clone tree figures. For 6 samples, we found convincing evidence of sample mislabeling, confirmed with the original authors. We provide both uncorrected and corrected tables in the repository associated with this paper (see Data availability).

We downloaded all available read accessions (284 clone tree samples in 6 clone trees, and 142 samples in 4 crosses) from the ENA. For each sample, we performed preprocessing as above (trimmomatic + rasusa) and then genotyped each sample with gramtools on the graph built from the 3,589 "analysis-set" samples above. To discover any missed variation (as these samples were not part of the 3,589 in the graph), or mutational events in progeny samples, we then ran all steps of our gramtools-based pipeline up to and including Gapfiller (S4 Fig).

By our evaluation pipeline standards, all samples were confidently resolved. We then aligned all pairs of progeny and parent samples (in the clone trees, aligning to the only parent, and in the crosses, aligning to both parents and looking for the closest parent (the parents

being usually highly diverged)) to infer any mutations. The only mutational events found were 2 SNPs, one in each DBLMSP1/2 gene, in 1 progeny sample of the cross HB3xDd2 (S27 Fig).

## Workflows and containerisation

The analysis steps underlying the results presented in this paper were implemented using bioinformatic workflows written in Snakemake [67], and versions of all software used were frozen in a Singularity container [68]. For access to these and all code and data underlying the results, see the Data availability section.

## Supporting information

**S1 Text. Text file supporting the supplementary figures of the paper.**
(DOCX)

**S1 Fig. Genomic context and protein domains in DBLMSP and DBLMSP2.** The 2 genes, marked with grey arrows, are located at a distance of 16.1 kbp from each other, inside an array of 8 contiguous genes spanning 32 kbp on chromosome 10. These genes are likely paralogs due to observed sequence sharing: All 8 have an N-terminal shared motif, a further 6 have a C-terminal SPAM domain, and DBLMSP and DBLMSP2 further share a DBL domain (domains shown as coloured circles below each gene). Figure annotated from a screenshot of the gene track for DBLMSP2 taken from PlasmoDB.
(TIF)

**S2 Fig. Geographical distribution of the 3,589 analysed *P. falciparum* samples.** A total of 29 countries are represented, with most samples located in the 2 regions with highest endemicity: sub-Saharan Africa and Southeast Asia. The base map comes from the freely distributed R package "maps," under a GPL-2 licence: https://cran.r-project.org/package=maps. The data and code to generate this Figure can be found at https://zenodo.org/doi/10.5281/zenodo.7677547.
(TIF)

**S3 Fig. Read statistics for the 3,589 analysed *P. falciparum* samples.** The upper panels show statistics measured on the reads directly: per-base quality (top-left panel) and read lengths (top-right panel). Per-base quality q gives the Illumina-estimated sequencing error rate $\epsilon$ as $\epsilon = 10^{\frac{-q}{10}}$. The lower panels show statistics measured after mapping the reads to the *P. falciparum* 3D7 reference genome: fragment length, estimated from the distance between paired-end reads (bottom-left) and fold-coverage, estimated from the number of reads at each mapped position. The data and code to generate this Figure can be found at https://zenodo.org/doi/10.5281/zenodo.7677547.
(TIF)

**S4 Fig. Existing and novel genotyping pipelines applied to the MalariaGEN data.** Panel (**a**) illustrates MalariaGEN's existing GATK-based pipeline, and panel (**b**) illustrates our new pipeline. Both first discover variants in each sample individually before regenotyping each sample at the union of all variants. GATK relies on the linear reference genome to do this, while gramtools uses a genome graph.
(TIF)

**S5 Fig. Framework used to evaluate the variant calls from the GATK and gramtools-based pipelines.** Starting from a tool's variant calls in a VCF file (middle), 2 independent evaluations were performed. First, for 14 samples with truth assemblies, the calls were directly compared

to the truth, by applying them to the 3D7 gene sequence and measuring edit distance for the whole gene (part a). Second, the calls were all applied to the reference genome, and the reads remapped to this induced reference. Incorrect or missing calls then appear from read pileups, as majority-differences compared to the reference base, coverage gaps, or inconsistent insert sizes between read pairs (part b).
(TIF)

**S6 Fig. Performance of the gramtools pipeline steps in DBLMSP and DBLMSP2.** The 2 panels, a and b, correspond to parts a and b of the evaluation framework in S5 Fig. Panel (**a**) shows the mean edit distance between the inferred gene sequence and the truth assembly for the 14 samples with truth assemblies (edit distance is scaled by gene length). Panel (**b**) shows the fraction of positions with pileup-based differences (top) and with low read coverage (bottom), after the sequencing reads are remapped to the 3D7 reference genome with each tool's called variants applied. A pileup-based difference is when the majority of reads disagree with the reference at a given position, given a minimum of 5 mapped reads, and low read coverage is defined as a position with fewer than 5 mapped reads. Each bar in panel b shows the mean across 500 of the 3,589 analysed samples. Across both panels, each coloured bar corresponds to one additional step in the gramtools-based pipeline, in the same order they are run (see S4 Fig). The "baseline" condition is not part of the pipeline and refers to using 3D7 reference gene sequence with no variants applied (in panel a: 3D7 sequence aligned to the truth assemblies; in panel b: sample reads aligned to the 3D7 reference genome). The data and code to generate this Figure can be found at https://zenodo.org/doi/10.5281/zenodo.7677547.
(TIF)

**S7 Fig. Global performance of the gramtools-based and GATK-based pipelines.** Panels a and b show the same metrics as S6 Fig. Metrics in panel b were computed on all 3,589 analysed samples. The data and code to generate this Figure can be found at https://zenodo.org/doi/10.5281/zenodo.7677547.
(TIF)

**S8 Fig. Frequency distributions of the evaluation metrics for the gramtools-based and GATK-based pipelines.** Each subplot shows the frequency distribution, across all 3,589 analysed samples, of the fraction of positions with pileup-based gaps (left-hand side plots) or differences (right-hand side plots), for DBLMSP (top) and DBLMSP2 (bottom). The mean is shown as a red vertical line (value shown in text next to it) and corresponds to the height of the coloured bars in S7 Fig panel b. The data and code to generate this Figure can be found at https://zenodo.org/doi/10.5281/zenodo.7677547.
(TIF)

**S9 Fig. Sequence filtering using pileup-based metrics.** In each panel ("baseline":no variant calling, "gram_joint_geno": gramtools-based pipeline, "malariaGEN": GATK-based pipeline), the total fraction of remaining gene sequences (out of the 3,589 analysed samples) passing filters is shown. Filters (colours) are applied in succession, on each set of remaining gene sequences, in the order they appear in the legend. The number of remaining sequences is given above each coloured bar. The data and code to generate this Figure can be found at https://zenodo.org/doi/10.5281/zenodo.7677547.
(TIF)

**S10 Fig. Peptides defined as shared are shared in many different countries.** The number of shared peptides by our definition (y-axis) that are found in both genes inside the same country, for up to 16 countries with high levels of sampling (defined as >50 available DBLMSP1/2

sequences; x-axis). A value of zero on the x-axis means the shared peptide is not found on both genes in any of these countries, and 16 means it is found on both genes in all of them. A majority (57%) of shared peptides are found in all of these countries, and 86% are found in at least 2 different countries, showing that the shared peptides are, overall, highly widespread geographically. The data and code to generate this Figure can be found at https://zenodo.org/doi/10.5281/zenodo.7677547.
(TIF)

**S11 Fig. Clustering tree with sequence sharing.** The 2 innermost rings show the gene and species of origin (as in Fig 2), and the outermost ring measures the level of sequence sharing between the 2 genes (see definition in the text). The data and code to generate this Figure can be found at https://zenodo.org/doi/10.5281/zenodo.7677547.
(TIF)

**S12 Fig. Shared peptide 10-mers are highly prevalent.** The frequency of shared peptides (y-axis), at each position (x-axis), is shown for the 16 countries with more than 50 sequences. Colour indicates frequency in each gene. Shared peptides are found at high frequencies, between 25% and 50%, across all countries. By extension, private peptides are also frequent, as any non-shared peptide is a private peptide. Values of zero, at the left and right ends of the x-axis, show the diverged flanks of the region, while values of one correspond to peptides that are always identical in both genes, i.e., where any mutations are likely eliminated by selection. On the left-hand side of the plots, DBLMSP2 displays a region with low shared peptide frequency across all countries, indicating this region has almost fully diverged between DBLMSP and DBLMSP2. The data and code to generate this Figure can be found at https://zenodo.org/doi/10.5281/zenodo.7677547.
(TIF)

**S13 Fig. Identification of *P. falciparum* orthologs in *Laverania* assemblies.** For each *P. falciparum* gene (panels), orthologs were searched for using minimap2 (preset: "-x asm20"). The y-axis shows the length of each hit normalised by the length of the *P. falciparum* gene sequence, and hits are coloured by % identity between query sequence and target in each *Laverania* assembly. The first 7 panels show genes occurring contiguously in a 40-kbp stretch of chromosome 10 on the *P. falciparum* 3D7 reference genome, and AMA1 was added as we expected it to be well conserved and found in single-copy. AMA1 could indeed be found in full length across all 6 *Laverania* assemblies, as was MSP11, a gene located in-between DBLMSP and DBLMSP2. We note that many genes are missing in *P. blacklocki*; this is most likely due to a restrictive form of whole-genome amplification prior to sequencing, which the original authors noted led to missing core genes in the resulting assembly [17]. The data and code to generate this Figure can be found at https://zenodo.org/doi/10.5281/zenodo.7677547.
(TIF)

**S14 Fig. Clustering tree of full-length DBLMSP1/2 sequences.** This Figure is the same as Fig 2, except that the tree was built from all unique DBLMSP1/2 full-length protein sequences, and not just of the DSR. While DBLMSP sequences from *P. reichenowi* and *P. billcollinsi* are outgroups in the clade of DBLMSP alleles, the sequences of DBLMSP and DBLMSP2 from *P. praefalciparum* fall nested within clades of *P. falciparum* alleles. This is consistent with a recent radiation of *P. falciparum* from a *P. praefalciparum*-like ancestor. DBLMSP2 is absent in *P. billcollinsi* and not shown in the tree for *P. reichenowi* as it is pseudogenised. The data and code to generate this Figure can be found at https://zenodo.org/doi/10.5281/zenodo.7677547.
(TIF)

**S15 Fig. Clustering tree of full-length AMA1 sequences.** As for S14 Fig above, the orthologous sequences from *P. praefalciparum* falls inside a *P. falciparum* clade, consistent with a recent radiation of *P. falciparum* from a *P. praefalciparum*-like ancestor, while orthologs from the other *Laverania* species occur as outgroups to *P. falciparum* alleles. The data and code to generate this Figure can be found at https://zenodo.org/doi/10.5281/zenodo.7677547.
(TIF)

**S16 Fig. HMM logos of private and shared DB sequences in the DSR.** One logo was produced for peptides found only in DBLMSP (top panel), only in DBLMSP2 (middle panel), and found on both genes (lower panel, labelled "Both"). The 3 tracks are broken into segments for visual clarity. At each position, observed amino acids are shown, with letter height proportional to amino acid frequency. In-between diverged N- and C-terminal regions, there is mostly 1 prototypical private sequence for each gene (first 2 tracks) and 1 prototypical shared sequence (or 2, in the C-terminal half of the protein domain). The data and code to generate this Figure can be found at https://zenodo.org/doi/10.5281/zenodo.7677547.
(TIF)

**S17 Fig. Number of distinct shared and private peptides per position.** For the 2 private and 1 shared MSAs, containing peptides only found in DBLMSP (top panel), only found in DBLMSP2 (middle panel), or both (lower panel), the total number of distinct peptide 10-mers at each position is shown. Mostly 1 to 4 peptides were observed at each position in the shared category, while 2 to 6 were observed on each gene only. This figure complements S16 Fig, which shows that mostly two 10-mer peptides occur in each gene with high frequency—here, total number is shown, regardless of frequency. The data and code to generate this Figure can be found at https://zenodo.org/doi/10.5281/zenodo.7677547.
(TIF)

**S18 Fig. Validation of MosaicAligner-inferred recombination events.** The blue dots show, for each target, its edit distance to the path of donors inferred by MosaicAligner (y-axis) and the edit distance to the single closest donor (x-axis). The grey dotted line shows y = x. All inferences reduce edit distances to the single closest donor, supporting adding recombination breakpoints to the alignment. The data and code to generate this Figure can be found at https://zenodo.org/doi/10.5281/zenodo.7677547.
(TIF)

**S19 Fig. Patterns of within- and interlineage recombinations.** The same clustering tree as in Fig 2 of the main text is shown, with the addition of dotted lines that connect 2 sequences if they were inferred to have recombined at some point in the past (see main text and Methods for how). Most recombination events occurred within the main lineages of the tree (e.g., within A, or within B.1), but a few events also occurred between highly diverged lineages of the tree (e.g., between C and A, or C and B.2). The data and code to generate this Figure can be found at https://zenodo.org/doi/10.5281/zenodo.7677547.
(TIF)

**S20 Fig. Specific examples of within and interlineage recombinations.** Five different recombinations are shown inside matrices, where, as in Fig 4 of the main text, each matrix depicts the mosaic alignment of 1 target sequence to the panel of 35 sequences. Sequences from DBLMSP (top) and DBLMSP2 (bottom) are separated by a white horizontal strip. Each cell is coloured by whether the size-10 peptide centred at that position occurs only in DBLMSP (blue-green), only in DBLMSP2 (orange), or in both (yellow). Recombinations mostly occur within the private DBLMSP2 lineage (all donors are mostly orange) and within the shared

DBLMSP2 lineage (all donors are mostly yellow). In the last panel, the target is a recombinant of a highly private and a highly shared sequence. The data and code to generate this Figure can be found at https://zenodo.org/doi/10.5281/zenodo.7677547.
(TIF)

**S21 Fig. Three mosaic alignments support gene conversion.** In each panel, 2 recombination breakpoints can be seen (red vertical lines). The target sequence is the fully opaque one (along its entire length; indicated with a black arrow), and the donor sequences (those the target aligns to) are shown as highlighted in places they match the target, and less opaque where they do not. In each panel, the target aligns to donors across the 2 different genes, consistent with gene conversion between the genes. The data and code to generate this Figure can be found at https://zenodo.org/doi/10.5281/zenodo.7677547.
(TIF)

**S22 Fig. Distribution of DBLMSP1/2 codon-level identity across 2,882 confidently resolved samples.** For all samples in which both DBLMSP1/2 sequences were confidently resolved, the DNA sequences of DBLMSP and DBLMSP2 within a single genome were aligned and the fraction of identical codons in the DSR recorded. Most samples have quite low identity levels (e.g., 0.2 up to 0.4), and a minority of samples have high identity levels, defined as >0.5 identity. The latter samples are illustrated in Fig 4 of the main text. The data and code to generate this Figure can be found at https://zenodo.org/doi/10.5281/zenodo.7677547.
(TIF)

**S23 Fig. Sequence motifs of samples from the 2 gene conversion events.** One logo was produced for each of the 2 conversion events in Fig 4 of the main text, and each logo split into 3 portions for visual clarity. While at many positions, the sequences in each conversion event overlap, each is enriched for different amino acids, and some positions have entirely different amino acids. This supports a distinct evolutionary trajectory for each event and thus at least 2 distinct gene conversion events having occurred in DBLMSP1/2. The data and code to generate this Figure can be found at https://zenodo.org/doi/10.5281/zenodo.7677547.
(TIF)

**S24 Fig. Geographical distribution of the 2 gene conversion events.** The 2 panels correspond to the 2 gene conversion events identified in Fig 4 of the main text (in the same order). In each panel, the number of samples in each geographical region is shown, both through the size and colour of dots. For both conversion events, samples are geographically widespread, occurring across west and east Africa and Southeast Asia. The base map comes from the freely distributed python package "plotly" (function "plotly.express.scatter_geo"), under an MIT licence: https://github.com/plotly/plotly.py. The data and code to generate this Figure can be found at https://zenodo.org/doi/10.5281/zenodo.7677547.
(TIF)

**S25 Fig. Identification of samples with putative CNVs.** For all 3,589 analysis-set samples, the mean and standard deviation (std) of the per-base read coverage of reads realigned the "induced reference" (S5 Fig, panel b) was measured in genes DBLMSP, DBLMSP2, and AMA1. For each gene, we produced a *coverage interval* {mean− 2 * std, mean + 2 * std}, which we consider a "plausible range" of gene-level coverage. The x-axis shows the ratio of the mean coverage in DBLMSP1/2 to that in AMA1, a gene that we assume to be single-copy in all samples. The marginal distribution histogram is shown on top. Most samples have a ratio of 1, and some have ratios <0.5 or >2, indicating possible copy-number changes. The y-axis shows the fraction of the DBLMSP or DBLMSP2 coverage interval overlapped by the AMA1 coverage

interval. Most samples have totally overlapping intervals (marginal distribution on right-hand side). Small overlap values indicate more likely true differences in coverage. Of the 6,123 analysed ("confidently resolved") DBLMSP and DBLMSP2 sequences, 31 had a fold-coverage >2 and an overlap <0.5 (bottom-right of plot), indicating putative duplication. Three of these overlapped with samples with evidence of gene conversion and were filtered out in that analysis (Fig 4 of the main text). The data and code to generate this Figure can be found at https://zenodo.org/doi/10.5281/zenodo.7677547.
(TIF)

**S26 Fig. Diversity and divergence levels in the DBL-spanning region (DSR) of DBLMSP1/2.** The 2 first panels measure, for each of DBLMSP and DBLMSP2, the percent codon identity of 2,882 randomly chosen gene pairs and is a measure of sequence diversity. The third panel shows the percent codon identity between DBLMSP and DBLMSP2 across all 2,882 samples where they were confidently resolved and is a measure of sequence divergence. Between-gene divergence exceeds within-gene diversity (lower codon identity across genes than within genes). The data and code to generate this Figure can be found at https://zenodo.org/doi/10.5281/zenodo.7677547.
(TIF)

**S27 Fig. Two SNPs identified in 1 genetic cross progeny.** In 1 progeny sample from genetic cross HB3xDd2, 2 SNPs were identified in DBLMSP1/2, one in each gene (panel a: DBLMSP2, panel b: DBLMSP). In both panels, the top track shows the parent gene sequence (HB3), and 4 subsequent tracks are shown below, each representing 1 different aligned sequence (grey horizontal bars). The first aligned sequence is the child sample gene sequence, showing a single SNP difference to the parent. To confirm these were spontaneous mutations and not single-base gene conversions from a homolog, 3 homologous sequences that could have been conversion donors were aligned to the parent: the orthologous sequence from the other cross parent (Dd2, second track), and the paralogous sequences from both parents (third and fourth tracks). No matches to these at the SNP positions can be seen. The data and code to generate this Figure can be found at https://zenodo.org/doi/10.5281/zenodo.7677547.
(TIF)

**S1 Table. Characteristics of the tools used in our new genotyping pipeline.** Each tool's approach and main strengths are summarised. "Specific" refers to low false-positive rates in variant calling, and "Sensitive" to high true-positive rates.
(DOCX)

## Acknowledgments

The authors thank Leah Roberts for reviewing the manuscript, Richard Pearson and Gavin Band for discussions of malaria genomics, and Richard Pearson for sharing MalariaGEN data ahead of the Pf7 release [69].

## Author Contributions

**Conceptualization:** Brice Letcher, Sorina Maciuca, Zamin Iqbal.

**Data curation:** Brice Letcher, Zamin Iqbal.

**Formal analysis:** Brice Letcher.

**Investigation:** Brice Letcher, Sorina Maciuca.

**Methodology:** Brice Letcher, Zamin Iqbal.

**Software:** Brice Letcher.

**Supervision:** Zamin Iqbal.

**Validation:** Brice Letcher.

**Visualization:** Brice Letcher.

**Writing – original draft:** Brice Letcher.

**Writing – review & editing:** Brice Letcher, Sorina Maciuca, Zamin Iqbal.

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
