## [Editor Report · Decision Letter 0]

30 Mar 2023

Dear Dr Letcher, 

Thank you for submitting your manuscript entitled "Gene conversion drives allelic dimorphism in two paralogous surface antigens of the malaria parasite P. falciparum" for consideration as a Research Article by PLOS Biology.

Your manuscript has now been evaluated by the PLOS Biology editorial staff, as well as by an academic editor with relevant expertise, and I'm writing to let you know that we would like to send your submission out for external peer review.

Once your full submission is complete, your paper will undergo a series of checks in preparation for peer review. After your manuscript has passed the checks it will be sent out for review. To provide the metadata for your submission, please Login to Editorial Manager (https://www.editorialmanager.com/pbiology) within two working days, i.e. by Apr 03 2023 11:59PM.

Kind regards,

Roli Roberts

Roland Roberts, PhD

Senior Editor

PLOS Biology

rroberts@plos.org

---

## [Decision Letter · Decision Letter 1]

18 May 2023

Dear Dr Letcher,

Thank you for your patience while your manuscript "Gene conversion drives allelic dimorphism in two paralogous surface antigens of the malaria parasite P. falciparum" was peer-reviewed at PLOS Biology. It has now been evaluated by the PLOS Biology editors, an Academic Editor with relevant expertise, and by three independent reviewers. 

You'll see that reviewer #1 is positive, but worries about availability of the sequence assemblies (please ensure that you are fully compliant with the PLOS data availability policy!), has semantic issues with the word “dimorphism” (and its relationship to paralogy – this also confused me first time round), and wants you to mark the mechanistic model as speculative and tone down the title. Reviewer #2 is also positive, but thinks that validation of the pipeline duplicates somewhat your previous paper and detracts from the main message (s/he suggests moving it to the supplement, leaving space for considering other stuff). Reviewer #3 is similarly positive, but wonders if you could have obtained the result simply by using the two haplotypes as two parallel reference samples, and asks about the timing with respect to the zoonotic jump from gorillas. The Academic Editor, in discussing these comments, said "R1 and R3 query the gorilla orthologs and R3 specifically asks for these to be included in the analysis which doesn't seem unreasonable in light of the authors' bottleneck hypothesis. I think this additional analysis would justify 'major revision'. R1 emphasises the usefulness of the method details but R2 wants them shifted to suppl. I think if the method has been published separately as R2 claims then the suppl is appropriate. I think providing the assemblies as requested by R1 is also appropriate."

In light of the reviews, which you will find at the end of this email, we would like to invite you to revise the work to thoroughly address the reviewers' reports.

Given the extent of revision needed, we cannot make a decision about publication until we have seen the revised manuscript and your response to the reviewers' comments. Your revised manuscript is likely to be sent for further evaluation by all or a subset of the reviewers.

**IMPORTANT - SUBMITTING YOUR REVISION**

*Re-submission Checklist*

*Published Peer Review*

*PLOS Data Policy*

*Blot and Gel Data Policy*

Sincerely,

Roli Roberts

Roland Roberts, PhD

Senior Editor

PLOS Biology

rroberts@plos.org

REVIEWERS' COMMENTS:

Reviewer #1:

This paper explores gene dimorphism in P. falciparum, by considering a pair of paralogous genes (DBLMSP and DBLMSP2) which were the subject of previous studies by some of the authors. Here' they propose a new genotyping pipeline, largely based on de novo assemblies, to obviate short-read mapping issues that occur in high-similarity paralogs. Using this method, they reconstruct the DBLMSP and DBLMSP2 sequences of several thousand samples including in the MalariaGEN dataset and, unsurprisingly, show that the pipeline is able to resolve these sequences better than the default GATK-based genotyping used by MalariaGEN. The authors then investigate sequence dimorphism in a domain of the two paralogs, and find that one of the "forms" is shared by the two genes. They also describe a number of recombination and conversion events, proposing a model for how dimorphism might have emerged as a result of population bottlenecks, speculating that this might have occurred when Pf ancestors jumped from gorilla to human hosts.

The paper is well-written and informative about the genetics of dimorphism, and the pipeline seems a valuable contribution. Generally, it would be of interest to those interested in parasite genetics, and suitable for publication. There are, however, some revision- including some conceptual adjustments- that I think are necessary.

1. Major Point: The primary output data from this analysis (namely, the thousands of reconstructed nucleotide sequences of DBLMSP and DBLMSP2) are not made available as far as I can see. I believe these data will be very useful to the research community. I appreciate that the authors have gone to some effort to make the pipeline available for reproducibility, but realistically this dataset could only be replicated from scratch with the resources of a major northern institution (e.g. EMBL-EBI), to the disadvantage of researchers in malaria-endemic countries. Please make a downloadable dataset of these sequences available, labelling them as they are labelled in Figure 6 (e.g. "PA1234-C DBLMSP2").

2. Important point: I feel the nomenclature and concept needs clarifying, in particular the concept of "dimorphism". The author insist that these genes have two, and exactly two forms; and then proceed to show a whole variety of different forms, 28 recombination breakpoints, gene conversions. So these domains are not dimorphic at all in the strict sense of the word, they're highly polymorphic, although each SNP within the domain appear to be dimorphic. It may be more correct to say that, for each locus, ancestries coalesce to exactly two individuals (to simplify, perhaps one could say that there are two major lineages that frequently interact with each other). I think the authors should spend more effort in the intro and discussion to clarify this.

3. Important point: Related to the above, one problem with using the DBLMSP and DBLMSP2 gene pair is that you're convolving "dimorphism" with paralogy. I can see that this gene pair was a logical choice in terms of showing the advantages of your pipeline, whose strength is primarily to resolve similar paralog sequences. But it does bring you into this rather confusing space where you're essentially analyzing the two DBs as a single entity (some sort of "pseudo-diploid"), and you're in fact showing that they are actually trimorphic (three clades in your Figure 6). I think a lot of readers may lose the plot at this point (the story would be a lot simpler had you picked other dimorphic genes). I believe you need to think how to guide the reader through these analyses.

4. Important point: The proposed model in the discussion is speculative (albeit plausible) and needs to be clearly marked as such. I am afraid that this also makes the manuscript title inappropriate. Even if the model is correct, and an additional form was generated by gene conversion, there is no explanation of how this form persists at ~50% prevalence, so to say that the conversion "drives" dimorphism does not seem correct. Also- the paper does not say whether the duplication leading to the paralogs occurred before or after the species jump. Is there evidence of this paralogy in gorillas?

Reviewer #2:

Letcher et al have undertaken a study in which they have employed a variant detection pipeline, gramtools, which uses genome graphs to assess variants that are called with a range of different genotype callers and is less affected by divergent haplotypes than other callers. After extensive validation of this pipeline they have applied it merozoite surface proteins in P. falciparum which appear to show highly diverged haplotypes, and are subject to frequent ectopic recombination. 

The improved variant calls suggest haplotypes are shared between the two MSPs, and further analysis of recombination patterns and phylogenetics strongly suggest this is a case of gene conversion. 

The work is extremely thorough, comprehensive and convincing, the results are notable in the context of malaria evolution. I have little hesitation in recommending it for publication. If I have a criticism to make, it is that the validation of gramtools in the content of Pf takes up a large part of the paper and does not show significant novelty beyond Letcher 21 / Hunt 22. This detracts from the more interesting story of gene conversion between MSPs and the paper might be better served by placing this in the supplementary. This may leave space for other evolutionary implications of this result (assuming they are not being dealt with elsewhere) such as the biological effects of these haplotypes and whether metrics such as KaKs etc indicate differing selective pressures on both private and shared haplotypes.

General nitpicking below: 

L113 - the range of tools appears to be cortex + octopus; these tools (and the reasons for choosing each) should be included

L127 / supplemental

Numbering and ordering of supplemental figures is either out of order or very confusing. 

L181 / fig s-2.1: without context it is hard to know how high this is - how does it compare to other paralogues that are not involved in cytoadhesion?

L256: Analysis appears sound, but it took me a few reads to figure it out, so perhaps some clarification that this is an alignment between genes is needed. 

L261 / Fig 5 do branch lengths derive from both the non-converted and converted regions? i.e. is there divergence between samples where gene conversion has occurred between DBs and if so how much?

L280: This figure is *very* complex and I'm not sure it adds much to the plot. 

L318: This section feels like a response to a reviewer question, but IMO doesn't add a lot to the paper. 

Reviewer #3: 

The authors have analyzed sequence reads from the MalariaGEN project (Plasmodium falciparum genome sequences from many thousands of samples). They have used a new pipeline they devised (and published in Ref.28) to characterize previously overlooked divergent allele sequences of two surface antigens (DBLMSP and DBLMSP2). I would not claim to fully understand the pipeline that leads to improved resolution of these sequences, but I think that they present convincing evidence of the efficacy of this method (Fig. 2). They then go on to show that there is evidence of recombination (probable gene conversion) between divergent sequences, within and between the two nearby loci. I found this very interesting.

Major comments

1. Given that it is already known that these two loci are dimorphic, would a standard assembly approach, run twice (using the two forms of each locus as reference) work just as well to resolve these sequences?

2. The authors note that Plasmodium falciparum arose by transmission of a gorilla parasite via a tight bottleneck, and discuss a model (Fig. 7) for the subsequent evolution of dimorphic loci through gene conversion between paralogous loci. So I was surprised that they did not include the DBLMSP and DBLMSP2 sequences from the gorilla parasite in their analysis; Otto et al. (2018; Ref.17) presented three genome sequences from this species.

Minor comments

1. Line 32, and later: I think the authors should avoid using "Pf" to denote P.falciparum.

Similarly, line 98 and later, I think the authors should avoid using "DBs". Finally, lne 144 and later, I think the authors should avoid using "MSA". Filling the paper with acronyms/abbreviations does not help its readability, and shortens the paper only marginally.

2. Figure 1. I did not find this helpful - I found it lacked sufficient detail/information to clarify anything beyond what is already stated in the text. 

3. Figure 2a and line 360: With two amino acids at a site in different alleles, the heterozygosity can rise to a maximum value of 0.5, when the two are at equal frequencies. In Figure 2a (right panels), for both genes there are numerous sites with values greater than 0.5, seemingly indicating that there are more than two alleles. Is this due to a number of rare amino acids at these sites?

4. Line 179, and lines 558 onwards: could clarify whether overlapping 10-mers are considered. (For example, if the 10-mer at sites 101-110 is considered, are those at 102-111, 103-112, etc., also considered?)

5. Figure 3. I did not find the explanation at lines 207-209 clear. Perhaps giving examples would help. Am I correct to think that the motif LRWFREWST, found in both genes near the bottom right, is a counterexample to what is stated in lines 207-209?

6. Line 285: Please give the number of unique protein sequences at this point.

7. Fig.6: I assume that the "subform" mentioned at line 314 is clade A.1. 

8. Line 443: explain MOI.

9. Line 461: 240kbp ? - which 240kbp ?

10. Line 463: "lied" should be "lay"

11. The numbering of the supplemental figures is odd.

---

## [Decision Letter · Decision Letter 2]

18 Dec 2023

Dear Dr Letcher,

Thank you for your patience while we considered your revised manuscript "Evolution of deeply-diverged lineages in two paralogous cell-surface antigens of the malaria parasite P. falciparum" for publication as a Research Article at PLOS Biology. This revised version of your manuscript has been evaluated by the PLOS Biology editors, the Academic Editor, and two of the original reviewers.

Based on the reviews, we are likely to accept this manuscript for publication, provided you satisfactorily address the remaining points raised by reviewer #3 and the following data and other policy-related requests.

IMPORTANT - please attend to the following:

a) We wonder whether you could make the Title more accessible and appealing? Maybe "Role for gene conversion in the allelic dimorphism of malaria parasite cell surface antigens" or "Evolution of deeply-diverged alleles of cell surface antigens of the malaria parasite" - happy to discuss this further by email.

b) Please address reviewer #3's remaining concerns.

c) Please address my Data Policy requests below; specifically, we need you to supply the numerical values underlying Figs 1AB, 2, 3AB, 4B, 5A, S3, S6AB, S7AB, S8AB, S9, S10, S11, S12, S13, S14, S15, S16, S17, S18, S19, S20, S22, S23, either as a supplementary data file or as a permanent DOI’d deposition. I note that you already have an associated GitHub deposition and a frozen Zenodo version thereof. Please could you confirm whether the aforementioned Figure panels can be generated using the data and code supplied in your Zenodo deposition, and if not, supply the underlying numerical values.

d) Please cite the location of the data clearly in all relevant main and supplementary Figure legends, e.g. “The data underlying this Figure can be found in S1 Data” or “The data underlying this Figure can be found in https://zenodo.org/record/8171279”

We expect to receive your revised manuscript within two weeks. 

*Published Peer Review History*

*Press*

Sincerely,

Roli Roberts

Roland Roberts, PhD

Senior Editor,

rroberts@plos.org,

PLOS Biology

DATA POLICY:

Regardless of the method selected, please ensure that you provide the individual numerical values that underlie the summary data displayed in the following figure panels as they are essential for readers to assess your analysis and to reproduce it: Figs 1AB, 2, 3AB, 4B, 5A, S3, S6AB, S7AB, S8AB, S9, S10, S11, S12, S13, S14, S15, S16, S17, S18, S19, S20, S22, S23. NOTE: the numerical data provided should include all replicates AND the way in which the plotted mean and errors were derived (it should not present only the mean/average values).

DATA NOT SHOWN?

REVIEWERS' COMMENTS:

Reviewer #1:

I have now reviewed the authors' replies and I am happy to accept the manuscript for publication.

Reviewer #3: 

I have the revised version, and find it greatly improved. I have only a few comments, which you will find below.

PBIOLOGY-D-23-00695R2

LETCHER et al.

Line 169: DBLMSP2 was not found in P. billcollinsi, even though an assembled genome is available. Laverania retain strong synteny, so it should be possible to comment whether the gene is indeed missing. Do the more divergent Laverania species have DBLMSP2? Or was it created by a duplication event after the divergence of the ancestor of P. billcollinsi?

Line 176: It would be strange if DBLMSP2 has retained a conserved function only in P. falciparum. That would imply that in both P. reichnowi and P. praefalciparum (and P. billcollinsi, if the duplication happened earlier) the gene was independently inactivated subsequent to divergence from the ancestor with P. falciparum; not very parsimonious to suggest the gene was required all along the specific lineage ;eading to P. falciparum (through various host species), but not in any divergent lineages.

Figure 2: The P. praefalciparum alleles are not easy to see. But having found them, it then seems remarkable that one within the B lineage is nested within a radiation of P. falciparum alleles. How can that happen?

Line 258-261: “The 209 samples …… supporting gene conversion occurring within each genome” sounds like the authors are suggesting 209 conversion events. Whereas, they are really invoking only two major events (line 263).

Line 290: I do not understand “we note that paralogous sequences diverge faster than orthologous sequences” ?

In general, at the interspecific level, that would not seem to make sense. Is this comment specific to intraspecific observations, reflecting limited recombination between paralogues?

Line 296: it is commented that the alleles in conversion cluster 1 do not cluster in the tree, with the excuse that the fraction of pasted sequence is lower. But the alleles in conversion cluster 2 do not cluster in the tree either.

Figure 5: For clarity: when I saw “Fraction of identical codons between paralogs” I first read this to mean the fraction of codons, within the converted region, that remain identical. Whereas I now believe this to be an indication of the length of the converted region.

Line 378: I think it should be “selected) in”

---

## [Editor Report · Decision Letter 3]

19 Jan 2024

Dear Dr Letcher,

Thank you for the submission of your revised Short Report "Role for gene conversion in the evolution of cell-surface antigens of the malaria parasite Plasmodium falciparum" for publication in PLOS Biology. On behalf of my colleagues and the Academic Editor, Michael Duffy, I'm pleased to say that we can in principle accept your manuscript for publication, provided you address any remaining formatting and reporting issues. These will be detailed in an email you should receive within 2-3 business days from our colleagues in the journal operations team; no action is required from you until then. Please note that we will not be able to formally accept your manuscript and schedule it for publication until you have completed any requested changes.

IMPORTANT:

a) I've taken two liberties with your manuscript. First, I've edited the genus name into the Title. Second, I've changed the article type to Short Report, which we think is more appropriate for the nature of the study (I meant to ask you to do this at the previous decision, but neglected to do so). The paper is already quite concise, so no reformatting is required.

b) Many thanks for clarifying the situation regarding your Zenodo deposition. However we need it to be cited in all relevant main and supplementary Figure legends, e.g. “The data and code needed to generate this Figure can be found in https://zenodo.org/records/XXXXXXX” (this may seem repetitive, but it serves to make the Figs more standalone. At a guess, the relevant Figs are Figs 1AB, 2, 3AB, 4B, 5A, S3, S6AB, S7AB, S8AB, S9, S10, S11, S12, S13, S14, S15, S16, S17, S18, S19, S20, S21, S22, S23, S25, S26). I have asked my colleagues to include this request with the aforementioned format-related requirements.

Sincerely, 

Roli Roberts

Senior Editor

PLOS Biology

rroberts@plos.org